# Surrounding Tissue Response to Surface-Treated Zirconia Implants

**DOI:** 10.3390/ma13010030

**Published:** 2019-12-19

**Authors:** Yohei Iinuma, Masatsugu Hirota, Tohru Hayakawa, Chikahiro Ohkubo

**Affiliations:** 1Department of Removable Prosthodontics, School of Dental Medicine, Tsurumi University, 2-1-3, Tsurumi, Yokohama, Kanagawa 230-8501, Japan; okubo-c@tsurumi-u.ac.jp; 2School of Dental Medicine, Tsurumi University, 2-1-3, Tsurumi, Yokohama, Kanagawa 230-8501, Japan; hirota-masatsugu@tsurumi-u.ac.jp (M.H.); hayakawa-t@tsurumi-u.ac.jp (T.H.)

**Keywords:** zirconia implant, large-grit sandblasting, acid etching, UV, soft tissue, collagen fiber

## Abstract

Yttria-stabilized tetragonal zirconia polycrystals (Y-TZP), which are partially stabilized zirconia, have been used for fabricating dental implants. This study investigated the soft tissue attachment, the collagen fiber orientation to zirconia at different surface conditions, and the bone response using implantation experiments in animals. The zirconia implant surfaces were treated with ultraviolet irradiation (UV), a combination of large-grit sandblasting and hydrofluoric acid etching (blastedHF), and a combination of blastedHF and UV (blastedHF+UV). The surface treated with blastedHF and blastedHF+UV appeared rough and hydrophilic. The surface treated with blastedHF+UV appeared to be superhydrophilic. Subsequently, tapered cylindrical zirconia implants were placed in the alveolar sockets of the maxillary molars of rats. The bone-to-implant contact ratio of blastedHF and blastedHF+UV implants was significantly higher than that of the non-treated controls and UV-treated implants. The four different surface-treated zirconia implants demonstrated tight soft tissue attachments. Perpendicularly oriented collagen fibers towards zirconia implants were more prominent in blastedHF and blastedHF+UV implants compared to the controls and UV-treated implants. The area of the soft tissue attachment was the greatest with the perpendicularly oriented collagen fibers of blastedHF+UV-treated implants. In conclusion, blastedHF+UV treatment could be beneficial for ensuring greater soft-tissue attachment for zirconia implants.

## 1. Introduction

Recently, yttria-stabilized tetragonal zirconia polycrystals (Y-TZP), which are partially stabilized zirconia, have been attracted as a promising alternative dental material to titanium [1,2,3,4,5,6]. Y-TZP has high mechanical strength including good flexural strength, fracture toughness, excellent esthetics owing to their white color, and no risk of allergy [7,8,9,10]. It is expected that Y-TZP can overcome the disadvantages of titanium implants such as their dark grayish color or metal sensitivity [11,12].

Most animal studies reported no significant difference in the bone responses between zirconia and titanium implants [13,14,15,16]. A few surface modifications to zirconia have been reported for the purpose of improving the bone-tissue response. For example, calcium ion incorporation [17], apatite coating [18,19], ultraviolet irradiation (UV) [20,21,22], sand-blasting [15,23,24], acid-etching [25], and laser treatment [26,27,28,29,30] have been postulated for the effective surface treatment of zirconia. A combination of large-grit sandblasting and acid etching was also reported to be a useful method [15,31,32,33]. Sandblasting and/or acid etching is available for commercially available products [6].

On the contrary, few studies have reported methods for improving the soft tissue attachment of zirconia implants. Soft tissue attachment is crucial for implants, since the soft tissue acts as an effective barrier that suppresses the access of microorganisms and their products [34]. The long-term prognosis of dental implants depends on the support provided by the soft tissue attachment to implants, which may aid early osseointegration [35]. The dimensions of the mucosal and connective tissue attachment are apparently similar for zirconia and titanium implants [36,37,38]. Animal experiments have demonstrated a higher degree of soft tissue integration around zirconia implants compared to that for titanium implants [39]. Collagen fiber bundles of the surrounding soft tissue are perpendicularly oriented to the natural tooth. Tetè et al. [40] observed the collagen fiber orientation around machined titanium and zirconia dental implants. They found that collagen fiber orientation, which was predominantly parallel or parallel-oblique, was similar for titanium and zirconia implants.

Some surface modifications to titanium implants reportedly improved the soft tissue attachment. Microgroove formation by laser treatment is a well-established technique [41,42,43,44,45]. Moreover, surface modification with titanium binding peptides, mesh structures, collagen nanofibers, and magnesium-doping were reported [46,47,48,49]. Raita et al. observed the perpendicularly oriented collagen fiber bundles in soft tissues surrounding titanium implant surfaces coated with collagen nanofibers [48]. Yoshihara et al. found that carbon deposition on a titanium surface was probably detrimental to bone formation, but effective in improving wound healing and soft tissue sealing around the surface of titanium implants [50]. However, to the best of our knowledge, little is known about the influence of the condition of the zirconia surface to soft tissue attachment. The aim of the present study was to investigate the soft tissue attachment and observe the collagen fiber orientation to zirconia with different surface conditions using animal implantation experiments. We expected that if surface modifications of zirconia implants will be able to fix the orientation of collagen fiber bundles to be perpendicular to the implant, the attachment of surrounding soft tissue will improve. As a result, bacterial invasion around zirconia implants will be prohibited and the probability of peri-implantitis will be reduced.

The different surface conditions included UV irradiation, a combination of large-grit sandblasting and hydrofluoric acid etching (blastedHF), and a combination of blastedHF and UV (blastedHF+UV). The bone response to surface modified zirconia implants was also evaluated.

## 2. Materials and Methods

### 2.1. Zirconia Materials

Disks and tapered cylinders were fabricated from Y-TZP (with 3 mol% yttria) (TZ-3YB-E; Tosoh, Tokyo, Japan). Y-TZP disks (12.0 mm in diameter and 1.0 mm in thickness) were used for characterizing the surface modifications. The disk surfaces were polished with #1200 waterproof paper and diamond particles (3, 1 and 0.25 µm) under running water. The disks were used for observing the surface morphology and measuring the contact angle against double-distilled water.

Tapered Y-TZP cylinders with constant conicity shape (0.35 mm in upper diameter, 0.61 mm in lower diameter, and 4.0 mm in length) used for animal experiments were fabricated using computer-aided design/computer-aided manufacturing (CAD/CAM) technique (Aadva Mill LD-I, GC Corp., Tokyo, Japan) as shown in Figure 1. Preliminary experiments were conducted for determining the design of tapered Y-TZP cylinders. Three differently tapered metal cylinders were used for assessing stability, after placing the implants into the sockets of extracted maxillary first molars in rats. Metal cylinder implants were casted from the plastic implant which was prepared by CAD/CAM technique based on Stereolithography (STL) data. After checking the stability or loosing of the metal implant one week after placement, the shape and size of the zirconia implant were determined according to the results of the metal implant placement. We fabricated the zirconia implants using the CAD/CAM technique, based on metal implant dimensions that were the most stable. The size and shape of zirconia implants were determined based on STL data.

The zirconia disk specimens were treated with UV, blastedHF, or blastedHF+UV treatment after polishing. Polished specimens were used as controls. UV specimens were prepared by UV irradiation with a UV ozone cleaner (BioForce Nanosciences, Inc., Salt Lake, UT, USA) at a power of 19 mW/cm^2^ for 20 min [51]. For blastedHF treatment, sandblasting was performed perpendicular to the zirconia surface at a distance of 20 mm with 180 µm alumina particles at 0.5 MPa air pressure, followed by acid etching of the blasted surface with 46% hydrofluoric acid (HF) for 5 min at room temperature [52]. After etching, the specimens were cleaned with an ultrasonic cleaner (VS-100III; AS ONE, Osaka, Japan) using ethanol and distilled water for 20 min. The blastedHF+UV specimen was prepared by blasting and acid etching, followed by UV treatment.

Tapered Y-TZP specimens were also treated with UV, blastedHF, or blastedHF+UV, with the same method as described above. As-machined specimens were used as controls for the tapered specimens.

### 2.2. Surface Analysis of Treated Zirconia

The surfaces of the control, UV, blastedHF, and blastedHF+UV specimens were first observed under a scanning electron microscope (SEM, SU1510, Hitachi High-Technologies, Tokyo, Japan) at an accelerating voltage of 15 kV after sputter coating with Au using an ion coater (QUICK COATER SC-701, Sanyu Electron, Tokyo, Japan). Both surfaces of the disks and tapered cylinders were evaluated. 

Examination with an atomic force microscope (AFM; Nanosurf Easyscan 2, Nanosurf, AG, Liestal, Switzerland) revealed the three-dimensional surface morphology and surface roughness (Sa) of the surface-modified disk specimens. AFM images were captured in air. The scans were obtained in the dynamic force mode with a tapping cantilever (Tap 190 Al-G, Budget Sensors; Innovative Solutions Bulgaria Ltd., Sofia, Bulgaria). AFM images were obtained for an area of 25 × 25 µm^2^. Surface roughness was measured as the three-dimensional arithmetic height (Sa) value obtained from the captured images on AFM analysis. Three specimens for each condition were measured.

Contact angles for each specimen with respect to double-distilled water were measured using a contact angle meter (DMe-201; Kyowa Interface Science Co. Ltd., Tokyo, Japan). The volume of the water drops was maintained at 1.0 µL, and 10 s measurements of each surface were made thrice under controlled conditions of 25 ± 1 °C and 46% humidity.

### 2.3. Animal Experiment for Zirconia Implants

This animal study was approval by the Animal Experimental Ethical Guideline of the Tsurumi University School of Dental Medicine (Certificate No. 29A034, 30A017, 19A015). A total of 16 male Wistar rats (180–200 g, 6 weeks old) were used in the study. We housed two rats in one cage in a temperature-controlled room at 20–25 °C with a 12 h alternating light–dark cycle and provided water and powdered feed ad libitum during the experimental period. In total, 8 cages were used for 16 rats. Each animal received one tapered cylindrical implant. A total of 16 implants were placed. Four control (as-machined), 4 UV, 4 blastedHF, and 4 blastedHF+UV zirconia implants were placed for three weeks. Control and blastedHF Y-TZP implants were sterilized using an autoclave before the animal experiments were conducted. UV irradiation to UV and blastedHF+UV Y-TZP implants was performed immediately prior to implantation.

The implants were inserted into the sockets of the extracted maxillary teeth of rats, as described by earlier reports [44,48]. Surgery was conducted under general anesthesia administered by an intraperitoneal injection of ketamine hydrochloride (47 mg/kg, Daiichi Sankyo Propharma Co., Ltd. Tokyo, japan) and medetomidine hydrochloride (0.4 mg/kg, Nippon Zenyaku Kogyo Co., Ltd. Fukushima Japan). The right maxillary first molar was extracted using forceps. After making an incision on the periodontal tissue, the sockets of the mesial roots of the right molars were enlarged using a dental reamer (#110, MANI, INC., Tochigi, Japan). A tapered Y-TZP cylindrical implant was placed within the prepared root with a press fit, and the antagonistic tooth was extracted to avoid the loading. After surgery, the rats were injected subcutaneously with benzyl penicillin G procaine (3,000,000 U/kg) and were awakened with an intraperitoneal injection of atipamezole hydrochloride (0.83 mg/kg, Nippon Zenyaku Kogyo Co., Ltd., Fukushima, Japan). The rats were euthanized 3 weeks after implantation by non-explosive inhalation of carbon dioxide gas and the tissues were harvested. Each implant site, including the implant and soft and hard tissues surrounding it was dissected using a diamond saw (Cutting Grinding System, BS-300CP band system; EXAKT, Apparatebau GmbH & Co., KG, Nordersted, Germany).

### 2.4. Histological and Histomorphometric Observation

The specimens were fixed in 10% neutral-buffered formalin solution for at least 7 days and were dehydrated with ethanol in increasing concentrations (70%, 80%, 90%, 96% and 100%). Subsequently, each specimen was embedded in methyl methacrylate resin. After polymerization, non-decalcified sections were prepared in the palatal–buccal direction using a cutting–grinding technique (Cutting Grinding system, BS-300CP band system and 400CS micro-grinding system, EXAKT, Apparatebau GmbH & Co., Norderstedt, Germany) [53]. The sections were prepared in a transverse direction, perpendicular to the axis of the implants and the thickness of the specimens was adjusted to approximately 70–80 µm by polishing with water-proof paper (#1200, #2000 and #4000) under running water using the EXAKT grinding system. Non-decalcified sections were stained with methylene blue and basic fuchsin and evaluated under a light microscope (magnification 200×) (BX51, Olympus Corp., Tokyo, Japan). In addition to assessing the soft-tissue response, bone responses to zirconia implants were also evaluated. The percentage of bone-implant-contact (BIC) was defined as the percentage of direct bone contact to the total length of the implant embedded in the maxillary bone and determined using an image analysis system (WinRoof, Visual System Division, Mitani Corp., Tokyo, Japan), as shown in Figure 2.

The interface between the soft tissue and implant was observed in order to evaluate the soft-tissue response to the zirconia implant. Birefringent collagen fiber bundles and their orientation within the soft tissue around the implant were observed under polarized light microscopy (ECLIPSE LV100N POL, Nikon Corp., Tokyo, Japan). Figure 3 shows the region of interest (ROI) for the evaluation of soft tissue integration. First, the length of the attachment of soft tissue to the zirconia implant surface (total length) was calculated. Subsequently, the length of the implant attached side of the attached area in which the collagen fibers were oriented perpendicularly to the zirconia implant (perpendicular-oriented length) was calculated. The soft tissue integration ratio was calculated as the ratio of the perpendicular oriented length to the total length. The area of the attached part in which the collagen fibers were oriented perpendicularly to the zirconia implant was also calculated. This value was denoted as the area of soft tissue integration. 

### 2.5. Statistical Analysis

The results of Sa, contact angle, BIC, and soft-tissue-integration ratio and area were analysed using one-way analysis of variance and Tukey’s post hoc test for multiple comparisons among means. Statistical analyses were conducted with Origin Pro 9.0 J (OriginLab Corp., Northampton, MA, USA). *p*-values of less than 0.05 were considered statistically significant.

## 3. Results

### 3.1. SEM and AFM Observations

Figure 4 and Figure 5 show the SEM images of the surface-modified zirconia disks and tapered cylinders. For zirconia disks, control showed a slight scratch due to polishing. The control and UV-treated specimens had smooth surfaces. blastedHF and blastedHF+UV surfaces had an entirely roughened texture. UV irradiation caused no changes to the surface morphology. The surfaces of the tapered cylindrical controls showed little damage due to machining. No clear difference was observed between the disk and cylinder specimens treated with blastedHF, UV, and blastedHF+UV.

AFM images also exhibited appearances similar to those observed on SEM, as shown in Figure 6. blastedHF- and blastedHF+UV-treated surfaces appeared rough.

### 3.2. Surface Roughness and Contact Angles

Table 1 shows the values for Sa and contact angles measured against double-distilled water. Sa values of blastedHF and blastedHF+UV are significantly higher those for the control and UV-treated implants (*p* < 0.05). No significant difference was observed in Sa values between control and UV, and blastedHF and blastedHF+UV implants (*p* < 0.05).

Significant differences were observed among the contact angles of the control, UV, blastedHF, and blastedHF+UV-treated surfaces (*p* < 0.05). The blastedHF+UV-treated surface was superhydrophilic.

### 3.3. Bone Response

Figure 7 shows the histology of bone formation around the four different zirconia implants three weeks after implantation. Failure or loosening of the zirconia implants did not occur during the preparation of histological samples, and severe inflammatory responses were not observed macroscopically in the soft and hard tissues surrounding the implants. There was a distinct gap between the implant and surrounding bone in the control and UV implants. On the contrary, tight connections were observed between the bone and blastedHF and blastedHF+UV implants. The results of BIC measurements are shown in Table 2.

The BIC values of the blastedHF and blastedHF+UV implants were significantly greater (*p* < 0.05) compared with those of control and UV implants (*p* < 0.05). No significant differences were observed in BIC values between control and UV, and blastedHF and blastedHF+UV implants (*p* > 0.05).

### 3.4. Soft-Tissue Attachment

Figure 8 shows the histological features of the soft tissue formed over the alveolar crest around the zirconia implants. There were no distinct differences among the soft-tissue response. Tight attachment of soft tissue to zirconia implant was observed.

Polarized microscopic images of the birefringent collagen fiber bundles of the gingival connective tissue formed around zirconia implants are shown in Figure 9. The images in Figure 9 are a higher magnification of the boxed areas of Figure 8. Polarized light images of natural teeth are also shown in Figure 9. Collagen fibers oriented perpendicular to the zirconia implants were more prominent and abundant in blastedHF and blastedHF+UV implants, compared to the control and UV-treated implants. Sparse perpendicular collagen fibers were observed in the control and UV zirconia implants. Longer and denser perpendicularly oriented collagen fibers were observed in blastedHF+UV-treated implants. The perpendicular orientation of collagen fiber bundles to blastedHF+UV was similar to that of natural teeth. The black-coloured substances in the boxed area of the blastedHF+UV specimen were presumed to be contaminations. The contaminations, which were generated by polishing, entered the gap between collagen fibers.

Table 3 lists the soft tissue integration ratio for the four different zirconia implants. UV, blastedHF, and blastedHF+UV implants showed significantly greater soft tissue integration values than those of the controls (*p* < 0.05). There were no significant differences among the soft tissue integration ratio of the UV, blastedHF, and blastedHF+UV implants (*p* > 0.05). The soft tissue integration area of blastedHF+UV was significantly higher than that of control, UV, and blastedHF implants (*p* < 0.05).

## 4. Discussion

The present study evaluated the soft tissue attachment and collagen fiber orientation to zirconia with different surface conditions by placing implants in the alveolar sockets of maxillary molars in rats. It found clear evidence of abundant perpendicularly oriented collagen fiber bundles in soft tissues around blastedHF+UV zirconia implants. blastedHF+UV zirconia implants also demonstrated a better bone response.

The zirconia implants were placed according to previously described methods [44,48]. In a previous study, screw-type or cylinder-type titanium implants were inserted into the sockets of the maxillary molars of rats. The cylinder-type titanium implant had a sharp tip for the initial fixation of the implant into the maxillary bone. It was very difficult to fabricate screw-type or cylinder-type implants with sharp tips in zirconia because of its excellent hardness. Thus, we designed tapered cylindrical Y-TZP implants after preliminary implantation experiments, which lacked sharp tips for initial fixation.

The effect of UV exposure to zirconia on bone response has already been reported [20]. It is well known that UV exposure to zirconia induces photocatalytic activity, which is similar to TiO_2_ [54]. UV produced a hydrophilic surface and removed and promoted the removal of the hydrocarbon layer from the zirconia surface [20]. blastedHF is also known to improve the bone tissue response [39]. blastedHF treatment produced greater hydrophilicity compared to UV treatment. A superhydrophilic zirconia surface was produced by blastedHF+UV treatment.

We first investigated the bone response to four different surface-treated zirconia implants. Generally, rougher surfaces promote greater and rapid bone formation [55]. The blastedHF and blastedHF+UV implants showed a significantly greater degree of BIC, as expected. However, UV treatment did not enhance the BIC values. The reason for this observation is unclear. Tapered zirconia implants used in this study lacked sharp tips for initial fixation. We assume that insufficient initial fixation led to BIC values for UV-treated implants. The relatively larger standard deviation for the BIC values of UV implants was also caused by insufficient initial fixation of the implants.

We observed the orientation of collagen fiber bundles for evaluating the soft-tissue attachment to the zirconia implant. The orientation of collagen fiber bundles between dental implants and natural teeth was associated with large differences in the soft-tissue response. Natural teeth exhibited perpendicularly oriented collagen fiber bundles, but dental implants showed parallel orientation of collagen fibers [40]. For example, Buser et al. [56] reported that collagen fibers in soft tissue were oriented parallel to the implant although a direct soft-tissue contact has been observed to smoothly polished sandblasted titanium surfaces. It is thought that the difference in the orientation of collagen fibers is one of the causes of peri-implantitis [57].

Normally, the transmucosal part of the implant of most implant systems is smooth or highly polished [58]. However, the present study revealed that surface modifications can influence the soft-tissue response to zirconia implants. Perpendicularly oriented collagen fiber bundles were clearly observed around the blastedHF+UV-treated implants. There are few reports about the presence of perpendicularly oriented collagen fiber bundles to zirconia implants. Yang et al. [59] reported that UV light exposure to roughened zirconia had a positive effect on human gingival fibroblast behavior, including cell adhesion, proliferation, and collagen release. They concluded that surface morphology and hydrophilicity are associated factors and both are governed by the behavior of human gingival fibroblasts. The present animal study also demonstrated that a roughened and superhydrophilic surface improved the orientation of collagen fiber bundles in soft tissue. Future studies should investigate the effect of surface roughness and/or hydrophilicity on soft-tissue attachment of zirconia implants in detail. For example, the oxygen plasma technique is also known to produce a superhydrophilic zirconia surface. Kobune et al. [60] reported that oxygen plasma treatment of zirconia promoted the initial attachment of oral keratinocytes by enhancing extracellular matrix components such as laminin γ2 better than UV treatment. Oxygen plasma treatment of blastedHF zirconia implants may provide different results.

Egawa et al. [61] reported that there were no significant differences in the initial attachment of periodontopathic bacteria between Y-TZP and titanium. It is pointed out that it would be very hard to clean the blastedHF+UV surface if the surface was contaminated after the exposure of the surface in the oral cavity. Schwarz et al. [62] assessed the efficacy of non-surgical therapy for the management of peri-implant diseases in a zirconia implant system. They tried two therapies. One was the combination of mechanical debridement and local antiseptic therapy using chlorohexidine digluconate and the other was Er:YAG laser therapy. They reported that both therapies were effective for short-time clinical improvements, but that a complete disease resolution was not achieved by these therapies. Yoshinari [63] suggested that modification with fluoride or antimicrobial peptide and/or atmospheric-pressure plasma irradiation will be useful for antimicrobial surface modifications of the zirconia implants. Further studies should be needed for cleaning the blastedHF+UV zirconia surface.

Clinically, blastedHF+UV treatment may possibly prevent peri-implantitis with zirconia implants, owing to the presence of perpendicular collagen fibers. The BIC values of blastedHF+UV were also acceptable. Thus, blastedHF+UV treatment would be useful for zirconia dental implants. It is expected that the presence of perpendicularly oriented collagen fibers produce tight adhesion of soft tissue to zirconia implant. The bonding strength of soft tissue to the blastedHF+UV zirconia implant will be further evaluated. Bacteria invasion is also another factor for peri-implantitis. The sealing ability of soft tissues surrounding blastedHF+UV will be also evaluated.

## 5. Conclusions

This study evaluated the soft-tissue response to different surface-treated zirconia implants using animal implantation experiments. Custom-designed zirconia implants were inserted into the maxillary alveolar sockets in rats. The bone response was also evaluated. The surfaces of blastedHF and blastedHF+UV appeared rough and hydrophilic. The surfaces of blastedHF+UV implants were superhydrophilic. Bone-to-implant contact ratios for blastedHF and blastedHF+UV implants were significantly greater than those of non-treated control and UV-treated implants. Perpendicularly oriented collagen fibers were more prominent in blastedHF and blastedHF+UV implants compared to the control and UV-treated implants. The area of soft-tissue attachment of the perpendicular collagen fibers was the (significantly) largest for blastedHF+UV implants. Thus, we can conclude that blastedHF+UV treatment may improve the soft tissue and bone responses and be useful for surface-treatment of zirconia implants.

## Figures and Tables

**Figure 1 materials-13-00030-f001:**
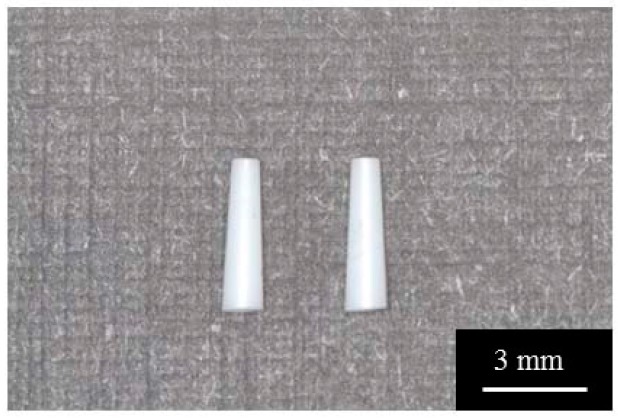
Tapered cylindrical zirconia implants fabricated by the CAD/CAM technique.

**Figure 2 materials-13-00030-f002:**
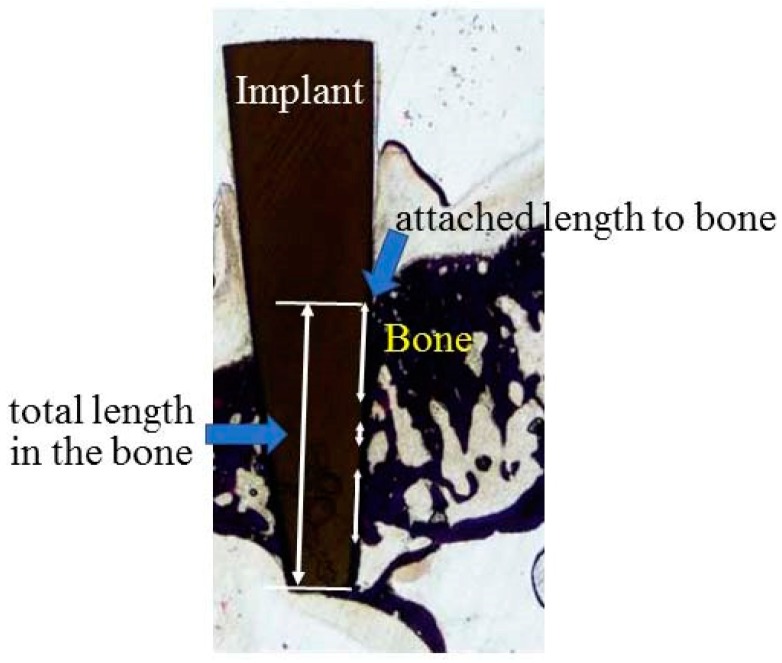
Schematic drawing for the measurement of the bone-implant-contact (BIC).

**Figure 3 materials-13-00030-f003:**
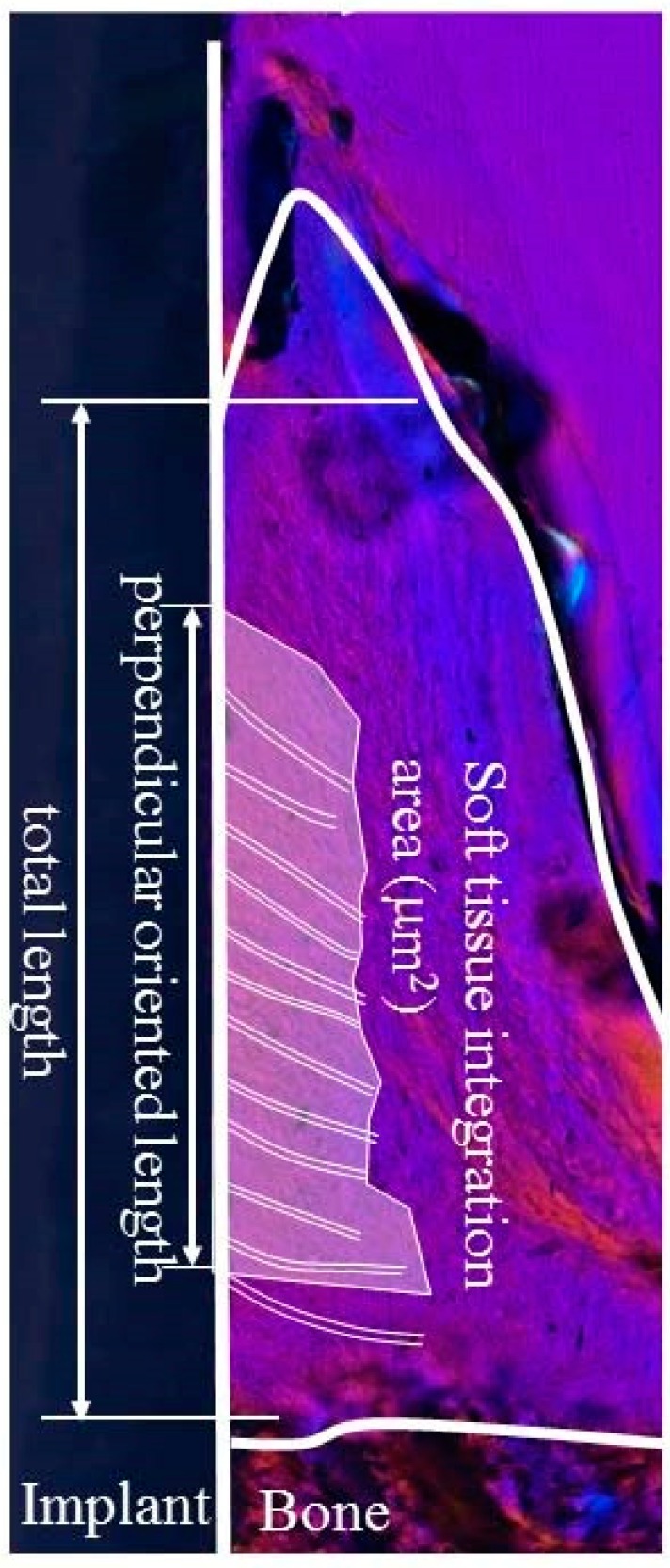
Schematic drawing of the region of interest (ROI) for quantitative analysis of perpendicularly oriented collagen fibers.

**Figure 4 materials-13-00030-f004:**
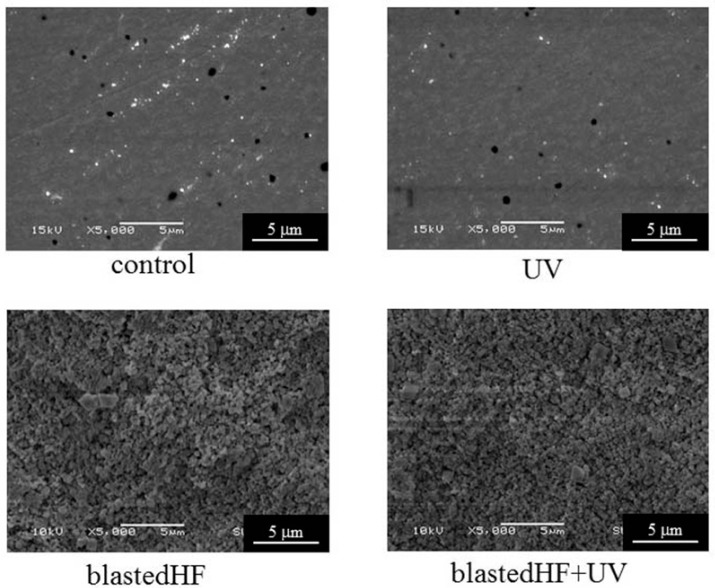
SEM images of surface-treated zirconia disks.

**Figure 5 materials-13-00030-f005:**
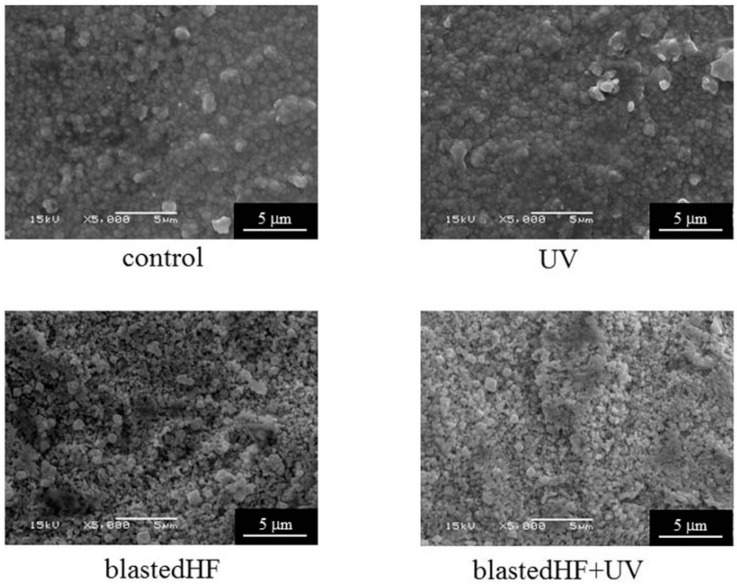
SEM pictures of surface treated tapered zirconia cylindrical implants.

**Figure 6 materials-13-00030-f006:**
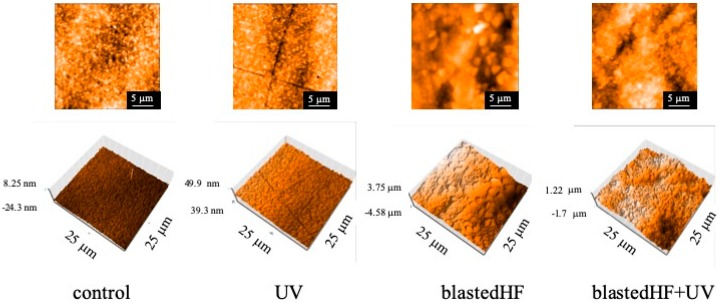
AFM images of surface treated zirconia disks.

**Figure 7 materials-13-00030-f007:**
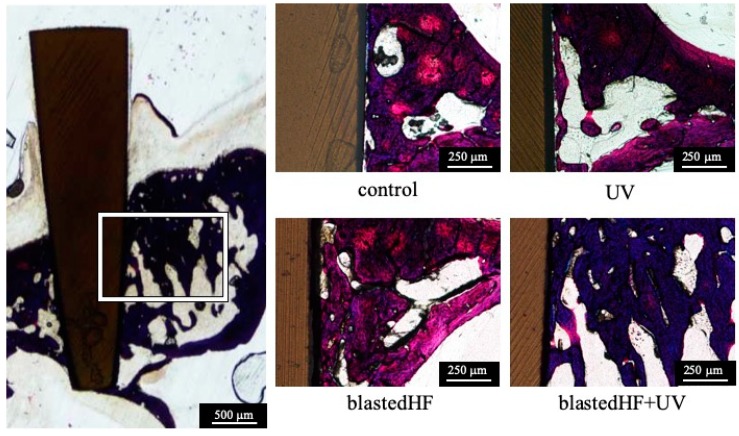
Histological features of the bone around zirconia implants. Each picture is a higher magnification of the boxed area.

**Figure 8 materials-13-00030-f008:**
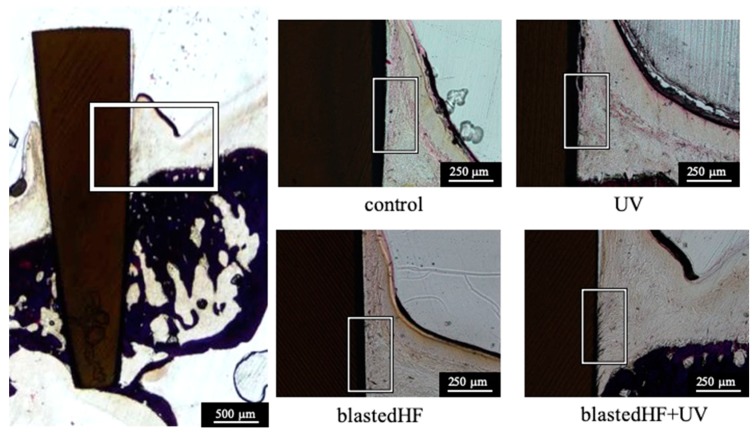
Histological features of the soft tissue around zirconia implants. Each picture is a higher magnification of the boxed area.

**Figure 9 materials-13-00030-f009:**
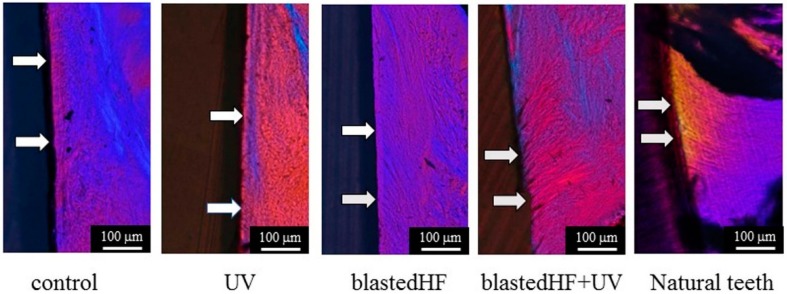
Polarized light microscopy of the soft tissue surrounding the zirconia implant and natural teeth. arrow: perpendicularly oriented collagen fibers.

**Table 1 materials-13-00030-t001:** Surface roughness (Sa) and contact angle (*θ*) of the specimens.

	Control	UV	blastedHF	blastedHF+UV
Sa (nm)	3.23 (1.14) ^a^	3.09 (3.65) ^a^	351.80 (42.44) ^b^	322.00 (41.91) ^b^
*θ* (°)	68.75 (2.91) ^a^	12.29 (2.92) ^b^	6.35 (1.18) ^c^	0.00 (0.00) ^d^

( ): Standard deviation; UV: ultraviolet irradiation; blastedHF: large-grit sandblasting and hydrofluoric acid etching; blastedHF+UV: blastedHF and UV; Different superscript letters indicate a statistically significant difference (*p* < 0.05).

**Table 2 materials-13-00030-t002:** Percentage of the measured BIC (%).

Control	UV	blastedHF	blastedHF+UV
58.10 (12.49) ^a^	50.32 (19.17) ^a^	80.39 (7.00) ^b^	85.87 (6.36) ^b^

( ): standard deviation; Different superscript letters indicate a significant difference (*p* < 0.05).

**Table 3 materials-13-00030-t003:** Soft-tissue integration of length and area.

	Control	UV	blastedHF	blastedHF+UV
Length (%)	23.24 (7.59) ^a^	38.05 (7.34) ^b^	41.72 (10.82) ^b^	48.87 (7.66) ^b^
Area (mm^2^)	13,455.51 (2680.50) ^a^	18,931.73 (1880.96) ^a^	26,508.26 (3059.00) ^a^	55,238.17 (8230.67) ^b^

( ): standard deviation. Different superscript letters indicate a significant difference (*p* < 0.05).

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
