# Peer review of "Surrounding Tissue Response to Surface-Treated Zirconia Implants"

_materials, 2019, doi:10.3390/ma13010030_

Round 1

Reviewer 1 Report

This study evaluates the surface characteristics of implants made of yttria-stabilized tetragonal zirconia polycrystals in vitro and in vivo in the animal model. The aim of the study is to compare different surface treatments with soft tissue integration.
The study evaluates an extremely interesting topic with very appropriate methods.
The results are clear and well presented. English language and style do not require corrections.
The study certainly represents an advancement of the knowledge of the field.
I strongly recommend the publication of this work.

Author Response

Thank you for your effort for reviewing the manuscript.

Reviewer 2 Report

Comments and suggestions for the Authors are reported in the attached file.

Author Response

Line 31: ‘’... have been used dental implant materials’’. The expression is not clear, since a word seems to be missed. Please check.

We changed the sentence as following.

Recently, yttria-stabilized tetragonal zirconia polycrystals (Y-TZP), which are partially-stabilized zirconia, have been attracted as a promising alternative dental material to titanium [1-6].

Line 51: The reference should be cited just after the names of the Authors of the study, i.e. Tetè et al. [ ]

We cited the reference just after the name.

Line 59: The use of the verb ‘’founded’’ in this context is not clear. Please check.

We changed “founded” to “found”. This is our mistake.

Lines 83-85: The dimensional accuracy of the manufactured specimens is not reported. Was it evaluated?

Metal cylinder implants were casted from the plastic implant which was fabricated by CAD/CAM technique based on STL data. The zirconia implant fabrication was preformed based on the SLT data in CAD/CAM technique. We added some sentences as following.

Preliminary experiments were conducted for determining the design of tapered Y-TZP cylinders. Three differently tapered metal cylinders were used for assessing stability, after placing the implants into the sockets of extracted maxillary first molars in rats. Metal cylinder implants were casted form the plastic implant which was fabricated by CAD/CAM technique based on Stereolithography (STL) data. After checking the stability or loosing of the metal implant one week after placement, the shape and size of the zirconia implant were determined according to results of the metal implant placement. We fabricated the zirconia implants using the CAD/CAM technique, based on metal implant dimensions which was the most stable. The size and shape of zirconia implants were determined based on SLT data.

Lines 114-115: It is not clear how the Authors have calculated the roughness. Was considered the arithmetic average of the measured values?

Present Sa value is obtained from the captured images on AFM analysis. Sample size for each measurement was three, and the average and standard deviation were calculated.

We modified the sentence as following.

Surface roughness was measured as the three-dimensional arithmetic height (Sa) value obtained from the captured images on AFM analysis. Three specimens for each condition were measured.

Line 124: ‘’..Two rats were housed...’’; It is not clear why this procedure was followed just for two rats. Please explain.

We housed two rats to one same cage. We changed the sentence as following.

We housed two rats to one same cage in a temperature-controlled room at 20-25 °C with a 12 h alternating light–dark cycle and provided water and powdered ad libitum during the experimental period. Totally, 8 cages were used for 16 rats.

Line 163: ‘’... the length of the attached area’’; the use of the term length is not clear, since the area is a derived quantity, while the length is a fundamental quantity. It is suggested to replace the word.

We changed the sentence as following.

Subsequently, the length of the implant attached side of the area in which the collagen fibers were oriented perpendicularly to the zirconia implant (perpendicular-oriented length) was calculated.

Line 169: The readability of Figure 2 should be improved, by enlarging it or increasing the font size.

We enlarged the figure size and increased the font size in Figures 2 and 3.

Lines 300-301: Do the Authors think that should be possible to improve the geometry of the zirconia implants to improve initial fixation?

We think it should be possible to improve initial fixation of implant if we can make screw-type zirconia implant or implant with sharp tip. Actually, first we tried to fabricate the screw-type zirconia implant by using CAD/CAM technique, it was very laborious but not impossible. Thus, we changed the design of the zirconia implant in the present study, Next time, we will try to make screw-type zirconia implant.

Line 246: ‘’.. These area are…’’. The subject of the sentence should be plural. Please check.

We changed sentence as following. Figure number was changed.

The images in Figure 9 are higher magnification of the boxed areas of Figure 8.

Lines 307, 314, 322: The reference should be cited just after the names of the Authors of the study.

We cited the reference just after the name.

Line 375: A capital letter is missing. Please check.

This is our mistake. We corrected it.

Reviewer 3 Report

The present study investigated the tissues response to zirconia implants treated with different surface conditions.

Although this is an important topic for dentistry, the present study has some issues and flaws that need to be corrected.

Comments

Introduction: It is well written and describes the objective of the study.

Material and methods:

Line 77: You used tapered Y-TZP cylinders (0.35 mm in upper diameter, 0.61 mm in lower diameter, and 4.0 mm). Is the conicity constant or no?

Line 81-84: It is not clear why did you use metal implantes and a week later you made new surgery for Zirconia implants placement . Could you explain?

Line 123: How did you determined the size of the sample?

Lines 133, 135, 137: Manufacturers, city and country.

Line 152: How did you adjust the thickness of the samples? Did you polish them?

Line 156: Could you explain how did you determined the percentage of bone/ implant contact and ilustrated the method with an image?

Author Response

Introduction: It is well written and describes the objective of the study.

Material and methods:

Line 77: You used tapered Y-TZP cylinders (0.35 mm in upper diameter, 0.61 mm in lower diameter, and 4.0 mm). Is the conicity constant or no?

Tapered Y-TZP cylinder ha a constant conicity shape. We changed as following

Tapered Y-TZP cylinders with constant conicity shape

Line 81-84: It is not clear why did you use metal implants and a week later you made new surgery for Zirconia implants placement. Could you explain?

Metal implants were used for deciding the shape and size of zirconia implants. First we made the metal implants with different size and implanted these metal implants into the sockets of extracted maxillary first molars in rats. We checked the stability or loosening of the metal implants and decided which size and shape of the metal implant was most stable. After deciding the shape and size of the implant, we fabricated the zirconia implants by CAD/CAM technique based on SLT data. We modified the sentence as following.

Preliminary experiments were conducted for determining the design of tapered Y-TZP cylinders. Three differently tapered metal cylinders were used for assessing stability, after placing the implants into the sockets of extracted maxillary first molars in rats. After checking the stability or loosing of the metal implant one week after placement, the shape and size of the zirconia implant were determined according to results of the metal implant placement. We fabricated the zirconia implants using the CAD/CAM technique, based on metal implant dimensions which was the most stable.

Line 123: How did you determined the size of the sample?

As described in line 132-135, one animal received one implant. Four conditions of surface modification, namely control, UV, blastedHF and blastedHF+UV were tested and four implants were used in each surface condition. Thus, 4x4=16 animals were used.

Lines 133, 135, 137: Manufacturers, city and country.

We added the manufactures, city and country as following.

Surgery was conducted under general anesthesia administered by intraperitoneal injection of ketamine hydrochloride (47 mg/kg, Daiichi Sankyo Propharma Co.,Ltd. Tokyo, japan) and medetomidine hydrochloride (0.4 mg/kg, Nippon Zenyaku Kogyo Co.,Ltd. Fukushima Japan). The right maxillary first molar was extracted using forceps. After making an incision on the periodontal tissue, the sockets of the mesial roots of the right molars were enlarged using a dental reamer (#110, MANI, INC., Tochigi, Japan). A tapered Y-TZP cylindrical implant was placed within the prepared root with a press fit. After surgery, the rats were injected subcutaneously with benzyl penicillin G procaine (3,000,000 U/kg) and were awakened with an intraperitoneal injection of atipamezole hydrochloride (0.83 mg/kg, Nippon Zenyaku Kogyo Co.,Ltd, Fukushima, Japan).

Line 152: How did you adjust the thickness of the samples? Did you polish them?

We used EXAKT cutting and grinding system for making non-decalcified section. Thinness of each sample was adjusted by polishing. We modified the sentence as following.

The sections were prepared in a transverse direction, perpendicular to the axis of the implants and the thickness of the specimens was adjusted to approximately 70-80 µm by polishing with water proof-paper (#1200, #2000 and #4000) under running water at EXAKT grinding system.

Line 156: Could you explain how did you determined the percentage of bone/ implant contact and illustrated the method with an image?

The percentage of bone/implant contact (BIC) was defined as the percentage of direct bone contact to the total length of the implant embedded in the maxillary bone. We modified the sentence as following, and added new figure (Figure 2) for the image of BIC measurement. As a result, figure number was changed one by one

The percentage of bone-implant-contact (BIC) was defined as the percentage of direct bone contact to the total length of the implant embedded in the maxillary bone and determined using an image analysis system (WinRoof, Visual System Division, Mitani Corp., Tokyo, Japan) as shown in Figure 2.

Reviewer 4 Report

Great manuscript

Please expand on your introduction and discussion on how your manuscript fits in the existing filed 

Author Response

Please expand on your introduction and discussion on how your manuscript fits in the existing filed.

We added some sentences in introduction and discussion section as following.

Introduction section

We expected that if surface modifications of zirconia implants will be able to control the orientation of collagen fiber bundles to perpendicular direction to the implant, attachment of surrounding soft tissue will improve. And as a result, bacterial invasion around zirconia implants will be prohibited and the probability of peri-implantitis will be reduced.

Discussion section

Normally, in transmucosal part of the implant of most implant systems is smooth or highly polished [58]. However, the present study revealed that surface modifications can influence the soft-tissue response to zirconia implants. Perpendicularly oriented collagen fiber bundles were clearly observed around the blatedHF+UV-treated implants. There are few reports about the presence of perpendicularly oriented collagen fiber bundles to zirconia implants.

              Clinically, blastedHF+UV treatment may possibly prevent peri-implantitis with zirconia implants, owing to the presence of perpendicular collagen fibers. The BIC values of blastedHF+UV were also acceptable. Thus, blastedHF+UV treatment would be useful for zirconia dental implants. It is expected that the presence of perpendicularly oriented collagen fiber produce tight adhesion of soft tissue to zirconia implant. The bonding strength of soft tissue to blastedHF+UV zirconia implant will be further evaluated. Bacteria invasion is also another factor for peri-implantitis. Sealing ability of soft tissues surrounding blastedHF+UV will be also evaluated.

Reviewer 5 Report

Congratulations to the authors for this very interesting study.

Overall, it looks okay. I have some concerns outlined below:

The implant design used in this study is not very good. When you look at the histological sections, it seems that a big portion of the implant was exposed to the oral cavity. This means that the implant must have come under some immediate loading. This should be discussed in more detail in the discussion section.

Figure 7: Blasted HF + UV - I have a query about the black coloured extensions from the implant in the boxed area. Are these artefacts from the specimen grinding. They don't look like collagen fibres (I might be wrong).

The authors have shown that blastedHF + UF surface promoted better and more perpendicular insertion/orientation of the collagen fibres. These are interesting results, but I have one concern. What if this surface exposed in the oral cavity? Although the good attachment might be beneficial in preventing inflammation, but if inflammation were to occur then this surface would be very hard to clean. This is something which should be taken into consideration and could possibly be elaborated in the discussion section.

Author Response

The implant design used in this study is not very good. When you look at the histological sections, it seems that a big portion of the implant was exposed to the oral cavity. This means that the implant must have come under some immediate loading. This should be discussed in more detail in the discussion section.

After the insertion of zirconia implant, we extracted the antagonistic tooth to avoid the loading. We forgot the description. This is our mistake. We added next sentence as following.

A tapered Y-TZP cylindrical implant was placed within the prepared root with a press fit, and antagonistic tooth was extracted to avoid the loading.

Figure 7: Blasted HF + UV - I have a query about the black coloured extensions from the implant in the boxed area. Are these artefacts from the specimen grinding. They don't look like collagen fibres (I might be wrong).

They are not the collagen fibers as you commented. They were contamination during the polishing.

We added the next sentence as following.

The black coloured substances in the boxed area of blastedHF+UV specimen were presumed to be contaminations. The contaminations which were generated by polishing entered the gap between collagen fibers.

The authors have shown that blastedHF + UF surface promoted better and more perpendicular insertion/orientation of the collagen fibres. These are interesting results, but I have one concern. What if this surface exposed in the oral cavity? Although the good attachment might be beneficial in preventing inflammation, but if inflammation were to occur then this surface would be very hard to clean. This is something which should be taken into consideration and could possibly be elaborated in the discussion section.

As you commented, the cleaning of the implant is a serious issue in dental clinics. We added the next sentence as following and added new references No.61-63.

Egawa et al [61] reported that there were no significant differences in the initial attachment of periodontopathic bacteria between Y-TZP and titanium. It is pointed out that it would be very hard to clan the blastedHF+UV surface if the surface was contaminated after the exposure of the surface in the oral cavity. Schwarz et al. [62] assessed the efficacy of non-surgical therapy for the management of peri-implant diseases at a zirconia implant system. They tried two therapies. One was the combination of mechanical debridement and local antiseptic therapy using chlorohexidine digluconate and the other was Er:YAG laser therapy. They reported that both therapies were effective for short-time cinnical improvements, but that a complete disease resolution was not achieved by these therapies. Yoshinari [63] suggested that modification with fluoride or antimicrobial peptide and/or atmospheric-pressure plasma irradiation will be useful for antimicrobial surface modifications of the zirconia implants. Further studies should be needed for cleaning the blastedHF+UV zirconia surface.

Egawa, M.; Miura, T.; Kato, T.; Saito, A.; Yoshinari, M. In vitro adherence of periodontopathic bacteria to zirconia and titanium surfaces. Dent Mater J 2013, 32(1), 101-106.

62.Schwarz, F.; John, G.; Hegewald, A.; Becker, J. Non-surgical treatment of peri-implant mucositis and peri-implantitis at zirconia implants: a prospective case series. J Clin Periodontol 2015, 42(8), 783-788, doi: 10.1111/jcpe.12439.

Yoshinari, M. Future prospects of zirconia for oral implants -A review. Dent Mater J 2019, doi: 10.4012/dmj.